# Category-Level 6D Object Pose Estimation in the Wild: A Semi-Supervised Learning Approach and A New Dataset

**Yang Fu**     **Xiaolong Wang**
UC San Diego

## Abstract

6D object pose estimation is one of the fundamental problems in computer vision and robotics research. While a lot of recent efforts have been made on generalizing pose estimation to novel object instances within the same category, namely category-level 6D pose estimation, it is still restricted in constrained environments given the limited number of annotated data. In this paper, we collect Wild6D, a new unlabeled RGBD object video dataset with diverse instances and backgrounds. We utilize this data to generalize category-level 6D object pose estimation in the wild with semi-supervised learning. We propose a new model, called **Re**ndering for **Po**se estimation network (**RePoNet**), that is jointly trained using the free ground-truths with the synthetic data, and a silhouette matching objective function on the real-world data. Without using any 3D annotations on real data, our method outperforms state-of-the-art methods on the previous dataset and our Wild6D test set (with manual annotations for evaluation) by a large margin. Project page with Wild6D data: https://oasisyang.github.io/semi-pose/.

## 1   Introduction

Estimating the 6D object pose is one of the core problems in computer vision and robotics. It predicts the full configurations of rotation, translation and size of a given object, which has wide applications including Virtual Reality (VR) [2], scene understanding [30], and  [42, 57, 31, 49]. There are two directions in 6D object pose estimation. One is performing instance-level 6D pose estimation, where a model is trained to estimate the pose of one exact instance with an existing 3D model [13, 34, 22, 50, 32, 5, 14]. However, learning instance-level model restricts its generalization ability to unseen objects. To achieve generalization to unseen instance, another direction is recently proposed to perform category-level 6D pose estimation using one model [48, 3, 41, 29, 6]. However, the large appearance and shape variance across instances largely increase the difficulty in learning.

To overcome this limitation, Wang *et al*. [48] take the initial step to collect real-world dataset and annotations for category-level 6D pose estimation. Combining synthetic data with free ground-truth annotations, they show the learned model can be generalized to unseen objects within the same category. While this result is encouraging, the generalization ability of the model is still limited by the number and the diversity of the data due to the challenges in annotating 6D object poses. Specifically, only 8,000 images across 13 scenes are collected and annotated from the dataset proposed in [48]. Thus it is still very challenging to generalize 6D pose estimation on diverse objects in complex scenes.

In this paper, we propose to generalize category-level 6D object pose estimation in the wild. To achieve this goal, we introduce a new dataset and a new semi-supervised learning approach. Our key insight is that, while annotating the 6D object pose is challenging, collecting the RGBD videos for these objects without labels is much easier and more affordable. On the other hand, there are infinite ground-truth annotations for synthetic data which come for free. We propose to leverage the benefits from both sides. We first collect a rich object-centric video dataset with diverse backgrounds and

36th Conference on Neural Information Processing Systems (NeurIPS 2022).

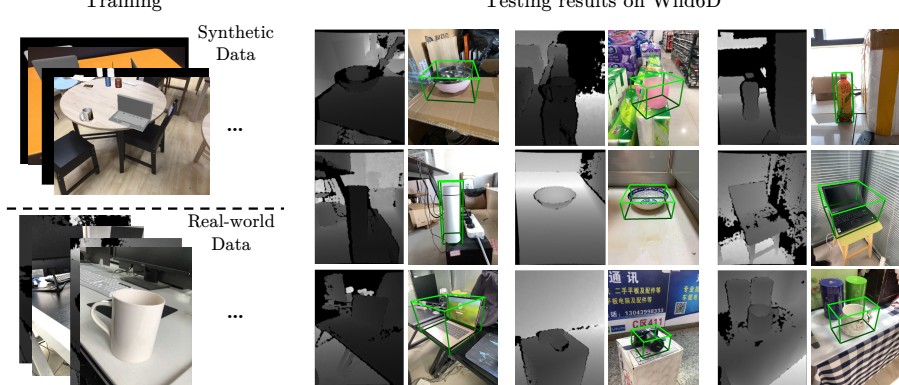

Figure 1: **Left**: we train our model with both synthetic data and real-world data under our proposed semi-supervised setting. **Right**: during inference, given the RGBD images, the object pose can be estimated precisely. Green bounding boxes show the 3D bounding boxes projection results on 2D images.

object instances using an RGBD camera. We train our model jointly using synthetic data with the free ground-truth 6D pose annotations and the unlabeled real-world RGBD videos via a silhouette matching objective. In this way, our pose estimation model can be generalized to in-the-wild data with minimum human labor. We collect an RGBD video dataset for 6D object pose estimation in the wild, namely *Wild6D*. Each video in the dataset shows multiple views of one or multiple objects (see examples in Fig. 1). In total, there are 5166 videos (> 1.1 million images) over 1722 object instances and 5 categories, which is significantly (300x) larger than the previous 6D object pose estimation dataset [48]. For evaluation, we annotate 486 videos over 162 objects.

Given this dataset, we design a novel model for semi-supervised learning, called **Re**ndering for **Po**se estimation **Net**work (RePoNet). The RePoNet is composed of two branches of networks with a *Pose Network* to estimate the 6D object pose and a *Shape Network* to estimate the 3D object shape. Given an RGBD image of an object, our Pose Network first estimates the Normalized Object Coordinate Space (NOCS) map [48], which will be integrated with lower-layer features to regress the object 9D pose (rotation, translation, and size parameters). Meanwhile, the Shape Network takes a category-level 3D shape prior as an input, and estimates the object shape for the current input instance. During training, we utilize both the synthetic data with the full ground-truths and the real-world data from Wild6D with foreground segmentation masks (obtained by Mask R-CNN [12]). Given the inputs with synthetic data, we apply the regression losses on both the NOCS map, 6D pose parameters, and object shape. Additionally, given the estimated pose and object shape, we perform differentiable rendering to obtain an object mask projected in 2D. A silhouette matching objective is proposed to compare the difference between the projected mask and the ground-truth mask. With the real RGBD data, although we do not have the 3D ground-truths, we can still learn with the silhouette matching objective by comparing the projected mask against the foreground segmentation. In this way, the gradients are back-propagated through the 6D pose and object shape to adjust both Pose Network and Shape Network for real RGBD data. During inference, we first apply our Pose Network to estimate the NOCS map with a given image. With the NOCS map output, the object pose can be computed by solving the Umeyama algorithm [43]. Some estimation results on Wild6D can be found in Fig. 1.

In our experiments, we first evaluate our approach with the dataset proposed in [48]. Our RePoNet shows a improvement over state-of-the-art approaches when trained in a fully-supervised setting (using 3D ground-truths from both real and synthetic data). By using the real-world data without its 3D ground-truth with semi-supervised learning, our performance can still be on par with previous approaches using full annotations. We then experiment with our Wild6D dataset and consistently show a large performance gain over the baselines on in-the-wild objects. We highlight our main contributions as follows:

- A new large-scale dataset Wild6D of object-centric RGBD videos in-the-wild for category-level 6D object pose estimation.

- A semi-supervised approach with RePoNet which leverages both synthetic data and real-world data without 3D ground-truths for category-level 6D object pose estimation.

- Our approach outperforms baselines by a large margin, especially when deployed on in-the-wild objects. We commit to **release our dataset, annotation and code** for benchmarking future research.

## 2   Related Work

**Category-level 6D Object Pose Estimation.** Recently, researchers have proposed to learn a single model for one category of objects instead of just one instance on pose estimation. For example, Wang *et al.* [48] propose a canonical shape representation of different objects called Normalized Object Coordinate Space (NOCS) to handle the instance variations. The object pose and size are calculated by the Umeyama algorithm [43] with predicted NOCS map and the observed points. Follow-up work using the NOCS representation has focused on improving the shape priors [41] and incorporating direct regression of object pose and size [3, 25, 6]. While these approaches can be deployed on unseen object instances, the generalization ability of these models is still limited by the scale and the diversity of real-world annotated data in the REAL275 dataset [48]. Only 13 scenes with 18 real objects in total are presented in REAL275. In contrast, we propose a semi-supervised learning approach with RePoNet, which leverages a new large-scale unlabeled dataset Wild6D for training. Both our method and the new dataset are the key components that lead to our goal of generalizing category-level 6D pose estimation in the wild.

**Large-scale 3D Object Datasets.** Various large-scale 3D datasets have been proposed for different tasks including reconstruction and pose estimation. For example, the Objectron dataset are collected and studied with large-scale object-centric videos and 3D annotations [1, 15, 26]. However, there are no depth images provided in Objectron, which might lead to ambiguities in 6D object pose estimation. Similarly, the recent proposed CO3D [37] dataset with diverse instances in different categories is also collected without recording the depth. While we can perform 3D reconstruction or generate the depth map using COLMAP [38], the error in predicted depth makes it difficult to be used for 6D pose estimation. Different from these datasets, the 3DScan [9] dataset records RGBD videos of different categories of objects. However, most objects are heavily occluded by hands or only partially visible with large objects like cars. The diversity of objects and scenes is also relatively low given limited human labor. Thus 3DScan is not suitable for pose estimation tasks. In this paper, we propose *Wild6D*, a dataset with RGBD videos containing diverse objects taken with diverse backgrounds. While the training set is not labeled, we provide 6D pose annotations for the test videos. To the best of our knowledge, Wild6D is the largest RGBD dataset for 6D object pose estimation in the wild.

**Rendering in Object Pose Estimation.** One effective refinement method for 6D object pose estimation is via rendering [24, 55, 20, 16]. For instance, Iwase *et al.* [16] propose to utilize differentiable rendering and learn the texture of a 3D model. However, it still depends on a pre-defined CAD model with a high-quality texture map for instance-level pose estimation and it is not feasible for category-level. Additionally, object pose estimation can be conducted via Analysis-by-Synthesis [44, 8, 52, 46, 40], but they usually require gradient descent for optimization leading to relatively slower inference speed. Critically, these works only focus on the instance-level setting and none of them can generalize to different instances. Inspired by recent work on 3D reconstruction [17, 23, 7, 18], our RePoNet learns the object shape and pose (including both NOCS map and R/T/S parameters) simultaneously using differentiable rendering to provide a loss function during learning [28], which allows the gradients to backprop through the whole network for training. While Manhardt *et al.* [29] also utilize differentiable rendering to improve category-level 6D pose, it is only applied in test time to adjust the pose, instead of training the network end-to-end. We show substantial improvement over all baselines training with RePoNet.

## 3   Wild6D Dataset

**Existing Category-level 6D Object Pose Datasets. NOCS** [48] is the most common dataset for category-level 6D object pose estimation, which consists of the synthetic CAMERA25 dataset and the real-world REAL275 dataset. The synthetic CAMERA25 dataset contains 300,000 RGBD images of 1,085 object instances from 6 categories for training and evaluation. The REAL275 dataset shares the same categories with CAMERA25 but is more challenging under the real-world background. It contains 4,300 RGBD images of 7 scenes for training and 2,750 images of 6 scenes for testing. Due to the limited scale and diversity of real data, the learned models from this dataset cannot generalize to in-the-wild scenarios. **Objectron** [1] is a large-scale dataset for object pose estimation and tracking.

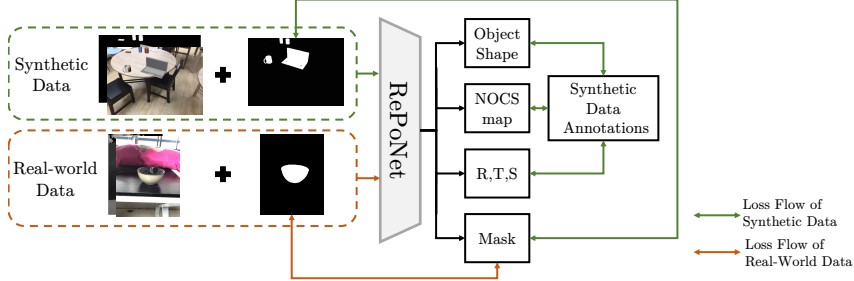

Figure 2: Our semi-supervised approach. For the synthetic data, we supervise it with all the annotations. While for the real-world data, we train it by comparing the binary mask generated by the rendering module with the object foreground segmentation.

| Datasets | RGBD | Real | #Categories | #instances | #images |
|---|---|---|---|---|---|
| Objectron [1] | | ✓ | 9 | 18K | 4M |
| CAMERA25 [48] | ✓ | | 6 | 184 | 300K |
| REAL275 [48] | ✓ | ✓ | 6 | 24 | 8k |
| *Wild6D* | ✓ | ✓ | 5 | 1.8K | 1M |

Table 1: **Comparison between Wild6D and existing datasets.** Wild6D significantly scale up the number of images, object instances, complexity of the scenes compared to previous RGBD datasets.

Different from NOCS [48], it consists of a collection of short object-centric video clips captured in the real world. However, no depth maps are provided in Objectron, which might lead to ambiguities in pose estimation.

**Wild6D Collection.** To achieve a categorical 6D pose for real objects, we collect a new large-scale RGBD dataset, named *Wild6D*. Each video in the Wild6D is recorded via the iPhone front camera showing multiple views of objects where RGB images and the corresponding depth images and point cloud are captured simultaneously. The videos are captured by different turkers with their own iPhones to guarantee the diversity of instances and background scenes. Three videos are taken for each object under different scenes. In total, Wild6D consists of 5,166 videos (>1.1 million images) over 1722 different object instances and 5 categories , *i.e.*, *bottle*, *bowl*, *camera*, *laptop*, and *mug*. Among this data, we split 486 videos of 162 instances to use them as the test set. Table 1 summarizes the statistics of Wild6D comparing previous datasets: Wild6D significantly improves the number of images, object instances, and scene complexity.

**Wild6D Annotation.** To annotate more than 10,000 images in the testing set efficiently, we propose a tracking-based annotation pipeline. Inside a video, we manually annotate the 6D object poses every 50 frames as keyframes. Given the annotation of the keyframe, we implement TEASER++ [51] together with colored ICP [33] to achieve the registration between the keyframe and the following frame and compute the transformation matrix. The ground-truth object pose of the following frame can be obtained by applying the transformation matrix to the keyframe annotation. Following this pipeline, we can obtain accurate ground-truths by only labeling around 5 keyframes per video.

## 4  Proposed Method

We propose the **Re**ndering for **Po**se estimation network (RePoNet) using both synthetic data and large-scale unlabeled real-world data in a semi-supervised manner.

**RePoNet overview.** The RePoNet is composed of two networks including the *Pose Network* to directly estimate the object 6D pose parameters (with the NOCS map as an intermediate representation) and the *Shape Network* to reconstruct the object shape. The outputs from both networks can go through a differentiable rendering [28] module which outputs a segmentation mask.

**Semi-supervised learning setting**. As shown in Fig. 2, we utilize two sets of data for semi-supervised learning: (i) synthetic data with full annotations including NOCS maps, CAD models, foreground segmentation, and 6D pose parameters, denoted as $\mathbf{D}_{syn}$; and (ii) a large-scale real-world RGBD dataset Wild6D with estimated foreground masks (using pre-trained Mask R-CNN [12]), denoted as $\mathbf{D}_{real}$. Both the synthetic data and the real-world data are utilized to train our RePoNet jointly. For the synthetic data, all the ground-truths are used as supervision signals for the *Pose Network* and the *Shape Network*. While for the real-world data, we directly feed the outputs from the *Pose Network* and the *Shape Network* into the differentiable rendering module to generate the binary mask,

then train the whole RePoNet by comparing the rendered mask with the object foreground mask (silhouette matching loss). While there can be a more delicate way to deal with the domain gap between simulation and real, we find the direct sharing of parameters during training already provides a simple yet effective solution for generalization.

One important contribution of this framework is to make the whole procedure with RePoNet differentiable, using the Shape Network, and a ConvNet to connect NOCS map to 6D pose parameters in the Pose Network. This allows the gradients to backprop through the network end-to-end. We will explain the architecture and objective details in the following subsections.

## 4.1 Framework Details

**Input data pre-processing.** Given an RGBD image, we first utilize the off-the-shelf Mask R-CNN [12] model to infer the segmentation mask for each object instance (See supplementary material for segmentation examples). We crop an object of interest from the RGB image and the corresponding depth map. The pixels in the depth map are back-projected as an observed point cloud. We denote the observation of an object instance as $X \in \mathbb{R}^{H \times W \times 3}$ for RGB channels and $P \in \mathbb{R}^{N_p \times 3}$ for the point clouds, where $W$, $H$ stands for cropped image resolution and $N_p$ is the number of sampled points.

**Feature extraction.** We employ the PSPNet [56] to extract the image feature as $f_x$ (Fig. 3 green box), and a PointNet [36] to extract the point cloud geometry feature as $f_p$ (Fig. 3 blue box). Meanwhile, for each category, we first define a category-level mesh prior $M \in \mathbb{R}^{N_v \times 3}$ to represent objects belong to a specific category, where $N_v$ is the number of vertices of the pre-defined mesh. We also use the PointNet to extract the shape prior feature as $f_{cate}$ (Fig. 3 purple box).

### 4.1.1 Pose Network.

Unlike instance-level 6D object pose estimation, where the CAD model of a specific instance is always given as the reference, the Pose Network is excepted to work well even without any instance models. To achieve this goal, we leverage the NOCS representation proposed in [48]. The architecture of the Pose Network is shown as the bottom part of Fig. 3. The Pose Network takes the RGB features $f_x$ extracted using PSPNet and the point cloud features $f_p$ extracted using PointNet as inputs. The two different modalities are combined as RGBD features using the dense fusion approach proposed in [45]. Specifically, the geometry feature $f_p^i$ of the $i$ th point is concatenated with its corresponding color feature $f_x^i$ and it is fed into a Graph Convolution Network (GCN) [27] (Fig. 3 yellow box) to obtain the point-wise RGBD feature, denote as $f_{rgbd}^i$.

Meanwhile, recall that we utilize a PointNet to obtain the point-wise geometry feature of the category-level shape prior $f_{cate}$ (on the Shape Network branch). The shape prior contains the prior knowledge of the category, *i.e.* coarse canonical shape and pose, but it does not align with the RGBD feature perfectly. To use the information of categorical shape prior and address the misalignment, we first apply a max-pooling operation on point dimension to obtain a global representation, then concatenate it with every point-wise RGBD feature, denoted as $f_{nocs}^i$, which can be used to estimate the NOCS coordinate. As the size and geometry of real-world object can be quite diverse, we introduce an **implicit function** $\Phi_{nocs}(\cdot)$ to deal with objects of the arbitrary shape (Fig. 3 gray box with MLP). Specifically, the $\Phi_{nocs}(\cdot)$ is implemented as a MLP with the point position and the corresponding feature as inputs, and predicts its NOCS coordinate as,

$$\Phi_{nocs}(f_{nocs}^i, p^i \in P) = p_{nocs}^i, \quad \forall i := 1 \dots N_p, \tag{1}$$

where P is the set of input point clouds and $N_p$ is the number of point clouds. The output is the NOCS coordinate $P_{nocs}^i$ for each point.

The NOCS map also explicitly reflects the geometric shape information of objects, similar properties are also shown with representations in [47]. Thus, the 6D pose, *i.e.*, rotation (**R**), translation (**T**), and scale (**S**) can be inferred from the NOCS map via a simple neural network. In detail, we directly take the RGBD feature $f_{rgbd}$ and concatenate it with the **intermediate results of NOCS coordinates**. This concatenated feature $f_{pose}$ will be used to predict the object pose **R**, **T** and **S** by a 3-layer ConvNet (Fig. 3 gray box with CNN). Using a neural network to connect the NOCS map and the 6D pose parameters makes the whole process differentiable, which is essential for training with the silhouette matching loss. More details of the network architecture can be found in the supplementary materials.

**Inference for 6D pose.** For 6D object pose estimation, only the NOCS map output of Pose Network is required for inference. The object pose can be computed by solving the Umeyama algorithm [43]

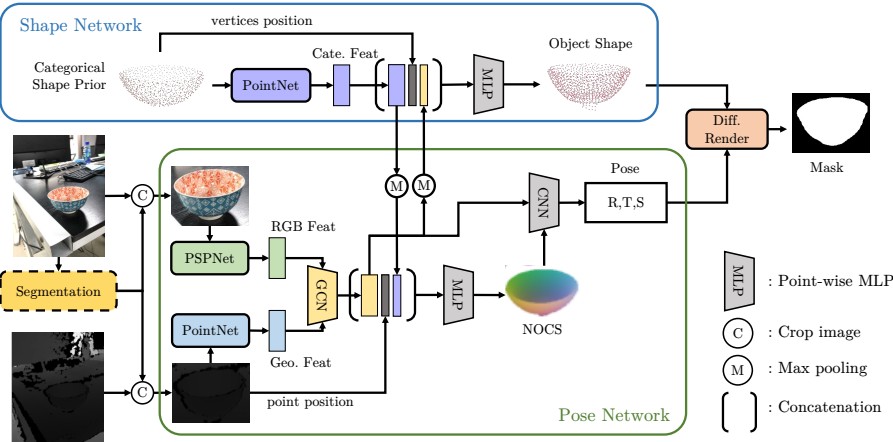

Figure 3: Overview of the proposed method. Given the input image and depth map, RePoNet estimates the object pose, NOCS map, and shape simultaneously via Pose Network and Shape Network. These two networks are bridged via the differentiable rendering module. By comparing the predicted binary mask with the input foreground mask, RePoNet can effectively leverage the real-world data without any annotations.

with the NOCS map. The 6D pose parameters $\mathbf{R}$, $\mathbf{T}$ and $\mathbf{S}$ are only used to perform differentiable rendering during training. The reason is that the directly estimated pose parameters from conv module is accurate when fitting the training data, but not as accurate as using NOCS map when generalizing to novel test instances.

### 4.1.2 Shape Network.

The Shape Network (top part of Fig. 3) aims to reconstruct the 3D shape of the input object from the given shape prior via mesh deformation. The major usage of this network is for performing differentiable rendering to provide training signals. The Shape Network is only used to estimate the object scale during inference time.

Given the categorical shape prior representation $f_{\text{cate}}$, the Shape Network borrows the point-wise RGBD feature of the input object $f_{\text{rgbd}}$ from Pose Network and concatenate them together, denoted as $f_{\text{shape}}$. Specifically, we max-pool the features $f_{\text{rgbd}}$ on point dimension and repeat it back with the number of shape prior points for concatenation with $f_{\text{cate}}$. We then predict the per-vertices deformation for the input mesh given $f_{\text{shape}}$. The deformation prediction network $\Phi_{\text{deform}}(\cdot)$ is also an implicit function taking $f_{\text{shape}}$ and mesh vertices positions M as the inputs, and outputs the 3D deformation as,

$$\Phi_{\text{deform}}(f_{\text{shape}}^i, \mathrm{m}^i \in \mathrm{M}) = \mathrm{m}_{\text{delta}}^i, \quad \forall i := 1 \dots N_{\text{v}}, \tag{2}$$

where $\mathrm{m}_{\text{delta}}^i$ represents the deformation for each vertex, $N_{\text{v}}$ is the number of the vertices, and $\mathrm{M}_{\text{delta}} = \{\mathrm{m}_{\text{delta}}^i\}_{i=1}^{N_{\text{v}}}$. Then the estimated mesh vertices of the given object is $\mathrm{M}_{\text{deform}} = \mathrm{M} + \mathrm{M}_{\text{delta}}$.

### 4.1.3 Differentiable rendering.

To obtain the supervision from 2D segmentation, we perform differentiable rendering using the outputs from the Shape Network and the Pose Network. Specifically, we feed the estimated object shape $\mathrm{M}_{\text{deform}}$ and estimated object pose $(\mathbf{R}, \mathbf{T}, \mathbf{S})$ into a differentiable rendering module [28] to generate the binary mask of the given object, as shown in Fig. 3. Given the rendering output binary mask, we can compare it against the Mask R-CNN segmentations and supervise the training of both pose network and shape network without additional 3D annotations.

### 4.2 Learning Objectives

We define multiple objectives for training the RePoNet, including the loss for 6D pose parameters, NOCS regression loss, shape reconstruction loss using synthetic ground-truths for supervision, and a silhouette matching loss which can be applied to both synthetic data and real data without 6D pose annotations. We introduce each loss as the following.

**Disentangled 6D Pose Loss.** Inspired by [24, 47, 39], we implement disentangled pose loss via individually supervising the rotation $\mathbf{R}$, translation $\mathbf{T}$ and scale $\mathbf{S}$. Instead of directly computing

parametric distances based on rotation matrix, we employ the variant of Point-Matching loss[1] [24, 20]. Additionally, we decouple the translation into the 2D location $(o_x, o_y)$ of the 3D centroid projection on the image plane and the object's distance to camera $t_z$. $(o_x, o_y)$ can be approximated as the bounding box center of the given object $(c_x, c_y)$. Given the camera intrinsics $K$, the translation can be calculated via Eq. 3,

$$\mathbf{T} = K^{-1}t_z[o_x, o_y, 1]^T \tag{3}$$

Therefore, given the ground-truth pose annotations $(\mathbf{R}^*, \mathbf{T}^*, \mathbf{S}^*)$, the objective function of 6D pose is formulated as Eq. 4,

$$\mathcal{L}_{\text{pose}} = \mathcal{L}_{\mathbf{R}} + \mathcal{L}_{\text{center}} + \mathcal{L}_z + \mathcal{L}_{\mathbf{S}} \tag{4}$$

Thereby,
$$\mathcal{L}_{\mathbf{R}} = \underset{\mathbf{x} \in M_{\text{CAD}}}{\text{avg}} \|\mathbf{R}\mathbf{x} - \mathbf{R}^*\mathbf{x}\|_2 \quad \mathcal{L}_{\text{center}} = \|o_x - o_x^*, o_y - o_y^* *\|_1 \quad \mathcal{L}_z = \|t_z - t_z^*\|_1 \quad \mathcal{L}_{\mathbf{S}} = \|\mathbf{S} - \mathbf{S}^*\|_1 \tag{5}$$

where $M_{\text{CAD}}$ is the ground-truth CAD model of the given object and $o_x^*, o_y^*, t_z^*$ can be computed via Eq. 3.

**NOCS Regression Loss.** To guarantee the pose estimation accuracy, we enforce the predicted NOCS coordinates $P_{\text{nocs}}$ to be closer to the ground-truth ones $P_{\text{nocs}}^*$. We use smooth-L1 loss [11],

$$\mathcal{L}_{\text{nocs}} = \sum_{x,y} \begin{cases} 0.5(x-y)^2/\beta, \text{if } |x-y| < \beta \\ |x-y| - 0.5 * \beta, \text{otherwise} \end{cases} \tag{6}$$

where $x \in P_{\text{nocs}}^*, y \in P_{\text{nocs}}$

**Reconstruction Loss.** Given the ground-truth CAD model $M_{\text{CAD}}$, the shape estimation is supervised by minimizing Chamfer Distance(CD) between $M_{\text{deform}}$ and $M_{\text{CAD}}$ via Eq. 7,

$$\mathcal{L}_{\text{recon}} = \sum_{x \in M_{\text{deform}}} \min_{y \in M_{\text{CAD}}} \|x - y\|_2^2 + \sum_{y \in M_{\text{CAD}}} \min_{x \in M_{\text{deform}}} \|x - y\|_2^2 \tag{7}$$

**Silhouette Matching Loss.** The above three objective functions, $\mathcal{L}_{\text{pose}}$, $\mathcal{L}_{\text{nocs}}$ and $\mathcal{L}_{\text{recon}}$, require the ground-truth information and cannot work with the real-world data without annotations. Thus, we further implement a silhouette matching objective function based on input foreground segmentation and the rendering results. We employ the negative IOU loss [54] to measure the difference between two masks and denote it as $\mathcal{L}_{\text{mask}}$.

Therefore, for the **fully-supervised training** with annotated synthetic data, the total loss function is described in Eq. 8, where the $\lambda$s are balance parameters.

$$\mathcal{L}_{\text{sup}} = \lambda_1 \mathcal{L}_{\text{pose}} + \lambda_2 \mathcal{L}_{\text{nocs}} + \lambda_3 \mathcal{L}_{\text{recon}} + \lambda_4 \mathcal{L}_{\text{mask}} \tag{8}$$

Meanwhile, under the **semi-supervised setting** with both annotated synthetic data and unlabeled data, the total loss function is then formulated as:

$$\mathcal{L}_{\text{semi}} = \mathbb{1}_{d \in \mathbf{D}_{\text{syn}}} (\lambda_1 \mathcal{L}_{\text{pose}} + \lambda_2 \mathcal{L}_{\text{nocs}} + \lambda_3 \mathcal{L}_{\text{recon}}) + \lambda_4 \mathcal{L}_{\text{mask}} \tag{9}$$

where $\mathbb{1}_{d \in \mathbf{D}_{\text{syn}}}$ is the indicator function.

## 5 Experiments

### 5.1 Implementation Details

**Semi-supervised setting.** We use the training data of CAMERA25 [48] along with the corresponding annotations and jointly train the model with images of REAL275 [48] or Wild6D without any 6D pose annotations. After cropping the object from the RGBD image, we first resize it to $192 \times 192$ and then randomly sample 1,024 points from both color image and depth map. To obtain the categorical shape prior, we choose a CAD model per category from the CAMERA25 training set manually and reduce its number of vertices to 1,024 as well. More configurations of RePoNet have been specified in supplementary materials. We adopt Adam [19] to optimize our model with the initial learning rate of 0.0001. The learning rate is halved every 10 epochs until convergence. We empirically set the balance parameters $\lambda_1, \lambda_2, \lambda_3$ and $\lambda_4$ to $0.2, 2.0, 5.0$ and $0.2$, respectively.

**Fully-supervised setting.** We share the same architecture and configurations with semi-supervised training but utilize all annotations of CAMERA25 [48] and REAL275 [48].

---

[1]We predict the quaternion representation of rotation matrix and follow the strategy in [50, 47] to deal with the symmetry of the object.

| NOCS map | Implicit function | $IOU_{0.5}$ | $IOU_{0.75}$ | 5 degree 5cm | 10 degree 5cm | $IOU_{0.5}$ | $IOU_{0.75}$ | 5 degree 5cm | 10 degree 5cm |
|---|---|---|---|---|---|---|---|---|---|
| | | **Fully supervised** | | | | **Semi-supervised** | | | |
| | | 77.0 | 55.7 | 35.4 | 63.2 | 71.7 | 47.9 | 22.1 | 47.5 |
| ✓ | | 79.1 | 59.1 | 36.9 | 65.3 | 74.6 | 49.9 | 29.7 | 57.1 |
| | ✓ | 79.2 | 60.0 | 39.8 | 66.6 | 72.2 | 47.7 | 24.3 | 51.6 |
| ✓ | ✓ | **81.1** | **63.7** | **40.4** | **68.8** | **76.0** | **52.2** | **33.9** | **63.0** |

Table 2: **Ablation on Implicit function and intermediate NOCS map.** We evaluate on REAL275 and the best results are highlighted in bold. w./w.o the nocs map refers to whether using the NOCS map as an intermediate result for pose regression. w./w.o. the implicit function stands for whether including the point/mesh coordinates in NOCS regression and shape reconstruction.

| silhouette loss | pose loss | $IOU_{0.5}$ | $IOU_{0.75}$ | 5 degree 5cm | 10 degree 5cm | $IOU_{0.5}$ | $IOU_{0.75}$ | 5 degree 5cm | 10 degree 5cm |
|---|---|---|---|---|---|---|---|---|---|
| | | **Fully supervised** | | | | **Semi-supervised** | | | |
| ✓ | | 78.1 | 61.9 | 39.7 | 65.2 | 74.8 | 47.1 | 27.9 | 59.7 |
| | ✓ | 76.7 | 58.2 | 30.7 | 59.7 | 73.1 | 46.0 | 18.7 | 47.9 |
| ✓ | ✓ | **81.1** | **63.7** | **40.4** | **68.8** | **76.0** | **52.2** | **33.9** | **63.0** |

Table 3: **Ablation study on pose regression loss and silhouette matching loss.** We evaluate on REAL275 and the best results are highlighted in bold.

## 5.2 Ablation Study

**Implicit function and NOCS map.** We first validate the effectiveness of two components in our approach: implicit function and intermediate NOCS map as described in Sec. 4.1.1. We evaluate the performance under fully-supervised setting and semi-supervised setting on REAL275 [48] and report the results in Table 2. Under the fully-supervised setting, by using the implicit function, the performance on $IOU_{0.5}$ and 5 degree, 5cm is improved by 2.2% and 4.4%, respectively. A similar improvement can also be observed for the model with an intermediate NOCS map. The best performance is achieved when combining the implicit function and NOCS map together. And the same conclusion can also be obtained under the semi-supervised setting as shown in Table 2.

**Pose Loss and Silhouette Matching Loss.** Besides the network architecture, we also show the effectiveness of the disentangled 6D pose loss and silhouette matching loss. The performance of the model w./w.o. the pose regression loss and w./w.o. the silhouette matching loss are listed in Table 3. Clearly, both pose loss and silhouette matching loss can enhance the object pose estimation performance under both fully-supervised and semi-supervised settings. For instance, the silhouette matching loss improves the $IOU_{0.75}$ and 10 degree 5cm by 5.5% and 9.1% compared with the model without it under the fully-supervised setting. This significant improvement shows the rendering module can effectively encourage communication between two networks, thereby boosting the pose estimation performance.

**Leveraging unlabeled real data.** Additionally, we report the performance of the proposed semi-supervised approach with training data from different sources in Table 4. As the baseline model, we first train it only with the synthetic data from CAMERA25 in a fully-supervised manner. Then, by using the unlabeled Wild3D RGBD videos, the performance is significantly improved by 6.8% and 5.4% on $IOU_{0.75}$ and 5 degree, 2cm. Similarly, the performance gains can be observed when adopting the REAL275 data without its annotations. Furthermore, with both REAL275 and Wild3D data, our model outperforms the baseline model by a large margin and approaches the fully-supervised model trained on REAL275 which can be served as the upper bound. More ablation studies about the amount of unlabeled real data used can be found in the supplementary materials.

**Amount of unlabeled real data.** We analyze the effect of using different fractions of unlabeled real data used during semi-supervised learning. We uniformly sample every 10% fraction of collected Wild6D training data for semi-supervised learning and evaluate the performance on REAL275 and Wild6D testing sets. As shown in Fig. 4, with more real data used during training, the object pose estimation performance is getting better.

## 5.3 Comparison with State-of-the-art Methods

**Performance on REAL275.** We split all existing methods into two groups by whether using full annotations of real data and compare their performance with RePoNet under both fully-supervised setting and semi-supervised setting, denoted as "RePoNet-sup" and "RePoNet-semi" respectively. As listed in Table 5, RePoNet-sup can achieve competitive results among all existing approaches under

| Methods | Training Data CAMERA25 | REAL275 | Wild6D | $IOU_{0.25}$ | $IOU_{0.5}$ | $IOU_{0.75}$ | 5 degree 2cm | 5 degree 5cm | 10 degree 2cm | 10 degree 5cm |
|---|---|---|---|---|---|---|---|---|---|---|
| RePoNet-semi | ✓ | | | 83.8 | 73.5 | 45.7 | 18.2 | 21.8 | 35.6 | 47.9 |
| | ✓ | | ✓ | 85.0 | 74.3 | 52.5 | 23.6 | 26.1 | 45.2 | 53.8 |
| | ✓ | ✓ | | 85.8 | 76.9 | 49.2 | 29.1 | 31.3 | 48.5 | 56.8 |
| | ✓ | ✓ | ✓ | 86.0 | 78.5 | 51.8 | 30.8 | 36.0 | 48.7 | 59.7 |
| RePoNet-sup | ✓ | ✓ | | 86.6 | 83.0 | 63.2 | 31.1 | 37.0 | 51.6 | 64.6 |

Table 4: **Ablation study of proposed semi-supervised method with different training data.** We evaluate the performance on REAL275 test set. "RePoNet-semi" and "RePoNet-sup" represent for the model trained w./w.o the 3D annotations of NOCS REAL275 respectively. Here we conduct experiments on five categories sharing by both REAL275 and Wild6D, *i.e.*bottle, bowl, mug, laptop and camera.

| Methods | $IOU_{0.5}$ | 5 degree 2cm | 5 degree 5cm | 10 degree 5cm |
|---|---|---|---|---|
| NOCS [48] | 78.0 | 7.2 | 10.0 | 25.2 |
| CASS [3] | 77.7 | – | 23.5 | 58.0 |
| Shape-Prior [41] | 77.3 | 19.3 | 21.4 | 54.1 |
| FS-Net [6] | **92.2** | – | 28.2 | 60.8 |
| DualPoseNet [25] | 79.8 | 29.3 | 35.9 | 66.8 |
| SGPA [4] | 80.1 | **35.9** | 39.6 | 70.7 |
| GPV-Pose [10] | 83.0 | 32.0 | **42.9** | **73.3** |
| RePoNet-sup | 81.1 | 35.1 | 40.4 | 68.8 |
| CPS++ [29] w/ ICP | 72.8 | – | 25.2 | ≤ 58.6 |
| SSC-6D [35] w/ ICP | 72.7 | 28.6 | 33.4 | 62.9 |
| CPPF [53] | 26.4 | – | 16.9 | 44.9 |
| UDA-COPE [21] | **82.6** | 30.4 | **34.8** | **66.0** |
| RePoNet-semi | 76.0 | **30.7** | 33.9 | 63.0 |

Table 5: **Comparison of our approach with the SOTA methods on REAL275**. Note that all SOTA methods listed in the top part are trained with full annotations of NOCS dataset including CAMERA25 and REAL275, while the bottom part is methods without using any real annotations. The best results are highlighted in bold.

the fully-supervised setting. Note our method is complimentary to the techniques proposed in SGPA, and our key contribution and focus lies on the semi-supervised counterpart for generalization. As shown in the bottom part of Table 5, the RePoNet-semi outperforms CPS++ [29] by a large margin under the semi-supervised setting [2]. Moreover, compared with the existing fully-supervised methods[3], RePoNet-semi achieves a comparable or even better performance without any 6D pose annotations. Additionally, our approach can perform shape reconstruction, but it's not our goal and we report its performance in the supplementary material.

**Performance on Wild6D.** Finally, we evaluate the performance of RePoNet on the Wild6D testing set as reported in Table 6. For some existing works, we directly test their pre-trained models trained on CAMERA75 along with REAL275. Since FS-Net [6] does not release the model, we cannot experiment on it. It can be observed that the pre-trained models cannot generalize to Wild6D due to the limited diversity of real data during training. For example, the performance of Shape-Prior [41] on three pose estimation metrics are all lower than 15%. On the other hand, the RePoNet with semi-supervised learn-

| Methods | $IOU_{0.5}$ | 5 degree 2cm | 5 degree 5cm | 10 degree 5cm |
|---|---|---|---|---|
| CASS [3] | 1.04 | 0 | 0 | 0 |
| Shape-Prior [41] | 32.5 | 2.6 | 3.5 | 13.9 |
| DualPoseNet [25] | 70.0 | 17.8 | 22.8 | 36.5 |
| GPV-Pose [10] | 67.8 | 14.1 | 21.5 | 41.1 |
| RePoNet-syn | 66.7 | 26.0 | 30.8 | 40.3 |
| RePoNet-semi | **70.3** | **29.5** | **34.4** | **42.5** |

Table 6: **Comparison of our approach with the SOTA methods on Wild6D.** "RePoNet-syn" is the model trained only on CAMERA75, while "RePoNet-semi" stands for the RePoNet trained on CAMERA75 and Wild6D. The best results are highlighted in bold.

ing on Wild6D achieves 29.5% and 34.4% on 5 degree, 2cm and 5 degree, 5cm, which is 10× better than Shape-Prior [41]. This significant improvement shows the better generalization ability of RePoNet. Also, the superior performance of RePoNet-semi over RePoNet-syn shows our proposed semi-supervised training can effectively leverage the in-the-wild data. We also show the qualitative results on Wild6D test set in Fig. 5.

---

[2]In Table 5, CPS++ only reports the performance under 10 degree, 10 cm.

[3]FS-Net and GPV-Pose are treated as the fully-supervised one, since it requires all real data and annotations.

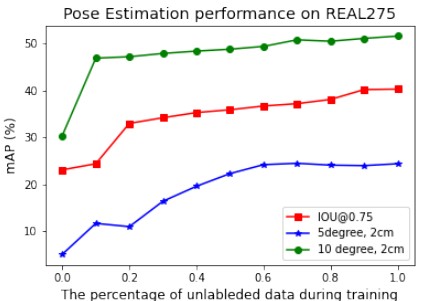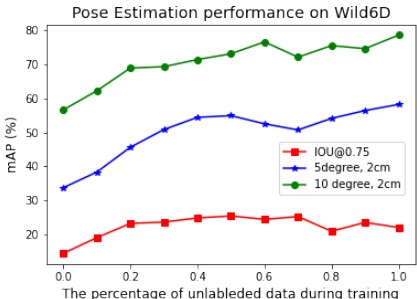

Figure 4: **Ablation study on semi-supervised training with different number of unlabeled real data**. Here, we only show the performance on bottle and evaluate it on both REAL275 and Wild6D dataset.

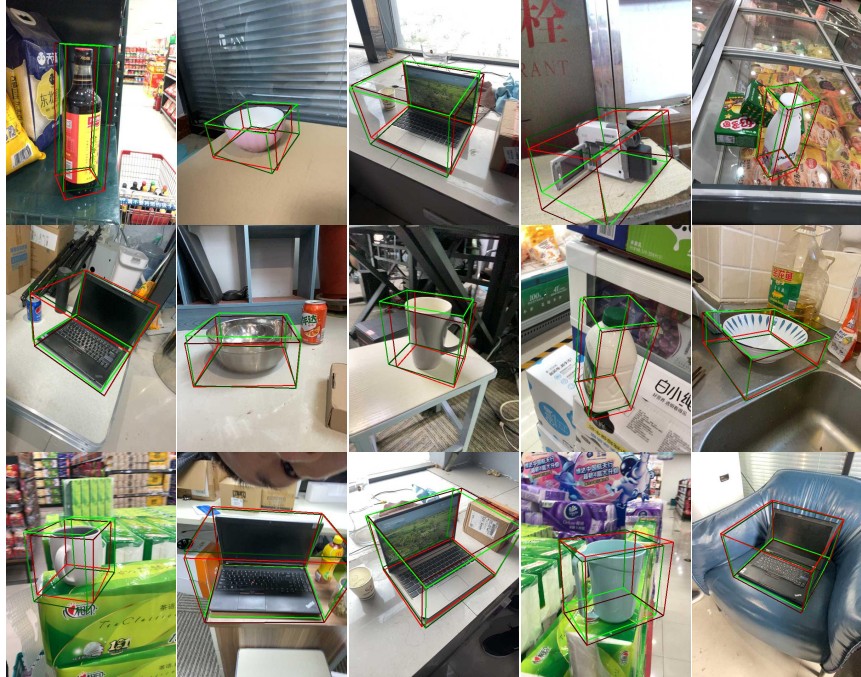

Figure 5: **Visualization Results on Wild6D test set**. Red 3D bounding boxes denote the ground truth, and the green boxes are estimation results via our proposed method.

## 6 Discussion

**Conclusion.** In this work, we consider the problem of generalizing 6D pose estimation to in-the-wild objects. Most of existing work are restricted in constrained environments due to limited number of annotated data. In an effort to resolve this limitation, we collect *Wild6D*, a new RGBD dataset with diverse instances and backgrounds. Instead of annotating them, we also propose RePoNet, a network that can leverage both the synthetic data and real-world RGBD data. Without using any 3D annotations on real-world data, RePoNet outperforms state-of-the-art methods on existing datasets and Wild6D test set by a large margin.

**Limitations.** Our proposed RePoNet may be hard to generalize to unseen categories. This is because RePoNet depends on the categorical mesh prior which is pre-defined for each category leading to the learnt deformation model fixed to a specific category. Alternative way to address this problem is to represent object shape via a deformation of a unit sphere to remove the category-level prior. Our dataset is currently limited to 5 categories to align with previous work for evaluation. We will extend it to more categories in the future.

**Acknowledgement.** This work was supported, in part, by grants from DARPA LwLL, NSF CCF-2112665 (TILOS), NSF 1730158 CI-New: Cognitive Hardware and Software Ecosystem Community Infrastructure (CHASE-CI), NSF ACI-1541349 CC*DNI Pacific Research Platform, and gifts from Meta, Google, Qualcomm.

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
