# OpenReview forum: "Category-Level 6D Object Pose Estimation in the Wild: A Semi-Supervised Learning Approach and A New Dataset"
_NeurIPS.cc/2022/Conference — NeurIPS 2022 Accept_

### Official Review · Reviewer_gzai · 2022-07-11

**Rating:** 6
**Confidence:** 5
**Soundness:** 3 good
**Presentation:** 3 good
**Contribution:** 3 good

**Summary:**

This paper presents a dataset and semi-supervised method for 6DoF object pose estimation with the aim of generalizing better. The first contribution is a dataset called Wild6D which is a dataset with 5166 RGBD videos of 1722 object instances from 5 categories. The second contribution is a neural network RePoNet that is trained to estimate 6DoF pose by training on both synthetic data (with full supervision) and real data (partial supervision). Experimental results indicate that the proposed method outperforms other methods on the 6DoF pose estimation task.

**Questions:**

It would be nice to see a discussion of the following points (in addition to the above limitations) in the rebuttal:

- How does Mask R-CNN quality affect the performance of the differentiable renderer?
- Could more categories be trained using the proposed approach?

**Limitations:**

The paper includes a short limitations section. It would have been nice to include a discussion of societal impacts -- this is missing currently.

**Strengths And Weaknesses:**

Overall, this paper has a nice contribution and I would recommend acceptance. In the following, I will list strengths, weaknesses, and other questions.

## Strengths

- The paper addresses an important problem of great practical importance. Generalizing 6DoF pose estimation is one of the grand challenges in 3D vision.
- The dataset contribution is important and can help make rapid progress in the community.
- The results are better than other state-of-the-art approaches.
- The idea of using full supervision from synthetic data and partial supervision from real data is obvious, but surprisingly no other paper has tried it as far as I am aware.

## Weaknesses

- Although the proposed dataset has lots of instances, it is limited to 5 categories. I presume this is to keep it consistent with the NOCS synthetic dataset. However, this limitation should be mentioned, in my view.
- While the architecture proposed in Figure 3 makes sense, there is not much justification for *why* this particular architecture was chosen over alternatives. For instance, why PSPNet was chosen? What is the rationale for combining RGB and geometric features in that way?
- The description of the Shape Network is missing details. It's not clear how the categorical prior is implemented. For instance, does it take a random vector as input?
- The results from Mask R-CNN can often be noisy, especially at the boundaries. It's not clear how this influences the result. It would have been nice to have a discussion on this.

---

> ### Author Response · Authors · 2022-08-02
> **Thank you and response to gzai**
>
> We thank the reviewer for the comments. We address each concern in the following. We have also highlighted the new contents in our revised paper with the color red.
>
> ---
>
> **Q1**: Although the proposed dataset has lots of instances, it is limited to 5 categories. I presume this is to keep it consistent with the NOCS synthetic dataset. However, this limitation should be mentioned, in my view.
>
> **A1**: Yes, we agree with the reviewer that the categories are selected to be aligned with NOCS for better comparisons. **We have added a statement for this in the limitation section of the revised paper.**
>
> ---
>
>
> **Q2**: why was PSPNet chosen? What is the rationale for combining RGB and geometric features in that way?
>
> **A2**: PSPNet is one of the most popular frameworks to obtain dense representation from RGB images. It is also applied in previous literature including Shape-Prior and CASS for the same purpose. We apply the same backbone for better comparisons. For combining the RGB feature and geometric feature, since both features are going to be projected to the 3D space for the following process, instead of regular convolutions, GCN provides an ideal manner to propagate the information efficiently among unorder 3D points and perform information fusion from two inputs.
>
> ---
>
>
> **Q3**: The description of the Shape Network is missing details. It's not clear how the categorical prior is implemented. For instance, does it take a random vector as input?
>
> **A3**: The categorical prior input for the Shape Network is a predefined category-level mesh, which has 1024 vertices. The Shape Network will output the deformation vector for each vertex of the mesh conditioned on the image input. We have added more details of Shape Network architecture **in the revised supplementary material**.
>
> ---
>
> **Q4**: The results from Mask R-CNN can often be noisy, especially at the boundaries. It's not clear how this influences the result. It would have been nice to discuss this.
>
> **A4**: While the Mask R-CNN results are not always perfect, given the object-centric Wild6D video dataset, the task becomes relatively simple and we can obtain high-quality segmentation in most cases. We have actually provided a discussion in our original supplementary material on Mask R-CNN and the segmentation result examples are shown in Figure 1 in our original supplementary material. We could further improve the segmentation results by using temporal consistency and tracking, but we find it unnecessary since the current masks are already satisfactory enough for providing training signals.
>
> ---
>
> **Q5**: How does Mask R-CNN quality affect the performance of the differentiable renderer.
>
> **A5**: Please refer to the **Q4** and **A4**.
>
> ---
>
> **Q6**: Could more categories be trained using the proposed approach
>
> **A6**: We believe our framework can be generalized to more categories than Wild6D, and we plan to extend the Wild6D with more categories and conduct more experiments on it.

---

> > ### Comment · Reviewer_gzai · 2022-08-08
> > **Thanks for the response**
> >
> > Thank you for responding to and addressing my questions and comments.

---

### Official Review · Reviewer_wqvk · 2022-07-12

**Rating:** 4
**Confidence:** 3
**Soundness:** 2 fair
**Presentation:** 2 fair
**Contribution:** 2 fair

**Summary:**

The paper proposes a new dataset for 6D pose estimation and a method named RePoNet, for estimating 6D object pose and size. The method falls in the category-level approach that predicts an object pose only based on its category.
In particular, the presented method is defined by two main branches a Pose Network and a Shape Network. The Pose Network uses the NOCS map as an intermediate representation to get the object pose, and the Shape Network uses the specific category 3D shape prior to leveraging the intra-class variation with the object mask.

The new dataset, named Wild6D, collects 5166 videos for five object categories under multiple views.
The method utilises the NOCS synthetic dataset, with all 6Dpose annotations, and the real dataset together with the proposed Wild6D to obtain masks and corresponding depth maps for training. The authors perform a detailed ablation analysis of the method on the datasets used. Results show that the proposed method outperforms the considered competing methods. Comparisons with state-of-the-art follow the metrics of [44].


**Questions:**

Please explain the implicit functions \Psi_{nocs}, and what is used to train the MLP.
Please explain \Psi_{deform}, how the accuracy of \Psi_{deform} is evaluated and the loss used.

Even if no qualitative results on the shape are provided,  line 320 reads ‘our approach can perform shape reconstruction’.
How relevant is the shape to generate the object bounding box? Why it is not used in inference (lines 224-225).

If the shape network is not used in inference, is the mask required? How is it used?
Please provide the parameters of the whole network and possibly runtime.

Please report correctly the comparisons with the semi-supervised methods using only real data, such as FS-Net.


**Limitations:**

Limitations are discussed, in particular, the difficulty to generalize.

**Strengths And Weaknesses:**

The paper poses the relevant question of mitigating the annotation effort in category-level 6D pose estimation. The authors propose to exploit a common strategy in many fields, based on training on synthetic images and resolving the domain gap with real images.
The authors refer to the NOCS dataset [44] (Camera25, with 300K  synthetic images, and Real275, with 8K real images). At the same time, the authors' contribution is to use only synthetic images from CAMERA25, with full annotations including  CAD models, segmentation and 6D pose parameters,  and taking from Real275 only the RGBD information (depth map and masks). Due to the domain gap between synthetic and real images, the authors claim that the method is semi-supervised.

The authors also contribute with a new annotated dataset, Wild6D, built from videos recorded with iPhone and a depth sensor. The Wild6D is annotated with 6D object poses every 50 frames, and the intermediate ones follow a tracking strategy.
The strength of the paper is in the competitive results on the 6D pose obtained by the semi-supervised method.

Major weaknesses of the paper are:
1. presentation.
2. The motivation of using only the synthetic parts with all the annotations.
3. A lack of comparison with most recent works and a discussion on them.

The description of the architecture is quite hard to follow as there are a huge amount of component networks, introduced via references, not always precise,  as in lines 207-209 ([43] does not even mention the NOCS, and it refers to a dense correspondence map, see also "Cdpn: Coordinates-based disentangled pose network for real-time RGB-based 6-dof object pose estimation").

The main difficulty in category-level 6D pose estimation is the intra-class object variation. As in previous work, the paper uses a 'shape prior' to face the problem. Adapting the shape-prior to the current object refers to a deformation network (supposedly as in [37]). No details on the accuracy of the deformation network and the loss used are provided, nor qualitative results.

The whole architecture, collecting several modules and networks, lacks substantial motivation.
Indeed,  the two losses,  the one for fully supervised training and the one for semi-supervised training, differ only in the data they collect. The balance parameters are the same, too, implying that there is no specificity in the semi-supervised learning for pose-sensitive features and geometric relationships.

The choice of training with the fully annotated synthetic images seems to be necessary for dealing with the prior shape. The motivation that synthetic data are at no cost is not convincing.

Recent works use only real data, following a different semi-supervised pattern to minimise the amount of training data.
For example, "GPV-Pose: Category-level Object Pose Estimation via Geometry-guided Point-wise Voting "(CVPR2022) uses just real data for training and uses for evaluation not only CAMERA25 and REAL275 but also LineMod (not considered by the paper). FS-Net [6] also uses only 'real data' for training, despite Table 5  treating it as fully supervised.

On the other hand, there are recent works relying only on synthetic datasets such as "CPPF: Towards Robust Category-Level 9D Pose Estimation in the Wild" (CVPR2022), "Self-Supervised Category-Level 6D Object Pose Estimation with Deep Implicit Shape Representation" (AAAI2022), "Category level object pose estimation via neural analysis-by-synthesis" (ECCV202), which the authors do not consider.

---

> ### Author Response · Authors · 2022-08-02
> **Thank you and response to wqvk (continue...)**
>
> **Q6**:On the other hand, recent works are relying only on synthetic datasets such as "CPPF: Towards Robust Category-Level 9D Pose Estimation in the Wild" (CVPR2022), "Self-Supervised Category-Level 6D Object Pose Estimation with Deep Implicit Shape Representation" (AAAI2022), "Category level object pose estimation via neural analysis-by-synthesis" (ECCV2020), which the authors do not consider.
>
> **A6**: We have included **all the results in the table below and our revision**. Our approach achieves much better performance compared to the mentioned papers. Note the ECCV 2020 paper did not report numbers but curves, thus we can only obtain the rough numbers from measuring the curves in the table below. But we are confident the results are far worse than our approach. We do not include the ECCV 2020 results in our revision given it is only roughly measured.
>
> | Method | IOU\@0.5 | 5cm, 5 degree| 10cm, 5 degree |
> |----------|--------------|----------|-------------|
> | ECCV 2020| -- | 10.0 | 13.5 | 17.5
> | AAAI 2022 | 73.0 |19.6 | | 54.5 |
> | AAAI 2022 w/ ICP | 72.7 | 33.4 | 62.9 |
> | CVPR 2022 | 26.4 | 16.9 | 44.9 |
> | Ours | **76.0** | **33.9** | **63.0**|
>
> ---
>
> **Q7**: Please explain the implicit functions $\Phi_{nocs}$, and what is used to train the MLP.
>
> **A7**: The details for $\Phi_{nocs}$ are explained in Line 197-201 before equation (1). The inputs include: (i) The concatenation of the RGBD feature $f_{rgbd}^i$ and the global shape prior feature (the max pooling values of $f_{cate}$ across all points), and (ii) the 3D position of each point of input RGBD image. The output is the NOCS coordinate corresponding to the input point. The MLP is trained end-to-end via back-propagation from the losses applied in the end, including disentangled 6D pose loss, silhouette matching loss, and NOCS regression loss (when ground truth is available).
>
> ---
>
> **Q8**:Please explain $\Phi_{deform}$, how the accuracy of $ \Phi_{deform}$ is evaluated, and the loss used.
>
> **A8**: The details for $\Phi_{nocs}$ are explained in Line 226-232 before equation (2). The inputs include: (i) The concatenation of the global RGBD feature (the max pooling values of $f_{rgbd}$ across all points) and the shape prior feature $f_{cate}^i$, and (ii) the 3D vertex position of each point on the shape prior. The output is the deformation applied on each vertex. The network $\Phi_{deform}$ is trained with the reconstruction loss (Equation (7)) and the silhouette matching loss.
>
> We have reported the 3D shape reconstruction performance on NOCS Real275 in our original supplementary material pdf. We kindly request the reviewer to look into Table 2 in our supplementary pdf.
>
> ---
>
> **Q9**: Even if no qualitative results on the shape are provided, line 320 reads ‘our approach can perform shape reconstruction. How relevant is the shape to generate the object bounding box?
>
> **A9**: We have reported the performance of object shape reconstruction on NOCS Real275 in our original supplementary material pdf Table 2. The shape reconstruction will provide the scale of the 3D bounding box which can be used as the object scale during inference.
>
> ---
>
> **Q10**: Why it is not used in inference (lines 224-225). If the Shape network is not used in inference, is the mask required? How is it used? Please provide the parameters of the whole network and possibly runtime.
>
> **A10**: The shape network is used, but not for rotation and translation estimation. **We have modified the sentence for avoiding confusion in our revised pdf**. The shape reconstruction is only used to compute the scale of the object, while for the 6D pose during inference, i.e. rotation and translation, we solve the Umeyama algorithm based on the estimated NOCS map and the depth map. The mask is required to segment the object during inference and we follow [3,39,46] to obtain masks via the off-shelf instance segmentation model. The total number of parameters is 22,635,313, the total size is 82.14MB and the runtime is around 10-15 FPS when testing on a single GTX 3090 GPU.
>
> ---
>
> **Q11**: Please report correctly the comparisons with the semi-supervised methods using only real data, such as FS-Net.
>
> **A11**: Please refer to the **Q5** and **A5**.

---

> ### Author Response · Authors · 2022-08-02
> **Thank you and response to wqvk**
>
> We thank the reviewer for the comments. We address each concern in the following. We have also highlighted the new contents in our revised paper with the color red.
>
> ---
>
> **Q1**: References are not always precise, as lines 207-209 [46] do not even mention the NOCS.
>
> **A1**: The NOCS map is widely utilized in category-level 6D pose estimation. While the method in [45] is for instance-level pose estimation, it has mentioned the NOCS paper multiple times such as “Similar to [46], we let the network predict a normalized representation of MXYZ.” in their paper. Thus, NOCS has indeed been mentioned and there are commons between the representations in [45] and NOCS. We believe it is correct to state both NOCS and the representations in [46] can reflect the geometric shape of the objects in line 207.
>
> To avoid confusion, we **change the sentence in 207 in our revised pdf** to “The NOCS map also explicitly reflects the geometric shape information of objects; similar properties are also shown with representations in [46].”
>
> ---
>
> **Q2**: No details on the accuracy of the deformation network and the loss used are provided, nor qualitative results.
>
> **A2**: We have provided the reconstruction loss in **Line 262-263 for the deformation network (Equation (7))**. The deformation network will also receive gradients back-propagated from other losses applied in the end, including reconstruction loss and silhouette matching loss. We have also reported the performance of object shape reconstruction on NOCS Real275 in our original supplementary material pdf. We kindly request the reviewer to look into Table 2 in our supplementary pdf.
> **We have also added additional reconstruction visualization in the revised supplementary materials.** Please look into Fig 3 in the supplementary material pdf.
>
> ---
>
> **Q3**: The whole architecture, collecting several modules and networks, lacks substantial motivation. Indeed, the two losses, the one for fully supervised training and the one for semi-supervised training, differ only in the data they collect. The balance parameters are the same, too…
>
> **A3**: We argue the simplicity of not over-tuning the hyperparameters on losses should be an advantage instead of a disadvantage. We use the same hyperparameter/weight for all losses, and still, show substantial improvement with our method. This actually shows the robustness of our approach instead of a lack of motivation. We could always tune the hyperparameters to obtain even better results, but our motivation is clear: We want to establish a simple and reproducible framework for generalizing 6D object pose in the wild. Paired with a new proposed dataset, we hope to allow more researchers to follow this study.
>
> ---
>
> **Q4**: The choice of training with the fully annotated synthetic images seems to be necessary for dealing with the prior shape. The motivation that synthetic data are at no cost is not convincing.
>
> **A4**: The cost of labeling synthetic data is far less than real data. It is common sense that synthetic labels are free in other 6D pose estimation papers, as well as general computer vision papers. The reason is that given the 3D models and data in the simulator, we can render the images and depth from different angles as many as we want automatically by writing a script. As the script is the one to generate the poses and cameras for rendering, thus this process comes with the pose and camera labels for free. Writing a script and computation for rendering are considered “free” in the context of comparing to manual labeling.
>
> ---
>
> **Q5**: Recent works use only real data, following a different semi-supervised pattern to minimize the amount of training data. For example, "GPV-Pose: Category-level Object Pose Estimation via Geometry-guided Point-wise Voting… FS-Net [6] also uses only 'real data' for training, despite Table 5 treating it as fully supervised.
>
> **A5**: As mentioned above, it is generally considered that real data annotations are the cost and synthetic labels are free. Thus we took FS-Net as fully supervised. **We have added a new note stating it only used real data for training in our revision**. For GPV-Pose, we would also like to remind the reviewer that the paper is just published in CVPR’22, after the NeurIPS submission deadline, which is considered a concurrent work according to the NeurIPS policy. Nevertheless, **we have included their results in our revision as well**. We would like to emphasize again that our method did not use any real data annotations, and it is expected that methods using real data annotations have more advantages.
>
> ---

---

### Official Review · Reviewer_rj61 · 2022-07-14

**Rating:** 7
**Confidence:** 4
**Soundness:** 4 excellent
**Presentation:** 4 excellent
**Contribution:** 3 good

**Summary:**

This dataset-and-method paper proposed a new unlabeled large-scale dataset for category 6D pose estimation in the wild, along with a new network design that can utilize that dataset in a weakly supervised manner. Most notably, the weakly supervision branch is on top of the desired shape and pose branches, thus allowing training both even without or with only a sporadic strong supervision signal. The proposed method is evaluated on existing datasets and existing methods are evaluated on the new dataset, showing that the proposed method exceeds state of the art.

**Questions:**

(1) Since iPhones were used, presumably with an own app to also capture depth, the rather accurate IMU could have been used to track the phone's movement (compared to the data-dependend models TEASER++/ICP, l.145).

(2) What is the estimated accuracy of the annotated poses, based on the errors introduced in labelling and tracking?

(3) Is the instance segmentation network pre-trained (and if yes, how / on which data), or trained along with the rest of the network in an end-to-end manner? For the latter case, I'd expect it to be rather unstable in the beginning of the training.

**Limitations:**

Yes, limitations and impact were appropriately addressed.

**Strengths And Weaknesses:**

+ The paper is well written and good to follow. All steps are well motivated and described, the network can likely be reproduced based on the detailed description and well-done figures.

+ The dataset acquisition is well thought off. Using "Turkers" to take videos with their own phones - compared to scientists - is very likely to have a reduced bias regarding scene selection, acquisiton etc.

+ The dataset is a significant improvement over prior datasets regarding size and variety. However, it is not fully annotated, thus potentially limiting its long-term impact.

+ The semi-supervised loss is well thought off, especially that it is based on the 6DoF pose which is the actual target (compared to a parallel branch / head in the network).

+ The proposed network and training protocol is well thought off. It is based on well established building blocks and combines them in a way that allows weakly supervised training. This allows it to efficiently leverage a large amount of unlabeled or weakly annotated real data.

+ The experiments are convincing and exhaustive. The proposed method is validated against prior art based on established datasets. Ablation studies show the impact of different building blocks. Some prior art methods are evaluated on the proposed dataset, forming a baseline evaluation.

+ The results are very good compared to prior art.

---

> ### Author Response · Authors · 2022-08-02
> **Thank you and response to rj61**
>
> We thank the reviewer for the comments. We address each concern in the following. We have also highlighted the new contents in our revised paper with the color red.
>
> ---
>
> **Q1**: The rather accurate IMU could have been used to track the phone's movement
>
> **A1**: In order to capture more accurate depth maps with higher resolution, we record all the videos via the front camera. At that point, the API provided by ARKit doesn’t support the camera pose estimation of the front camera. Thus we were not able to track the phone's movement.
>
> ---
>
> **Q2**: What is the estimated accuracy of the annotated poses, based on the errors introduced in labeling and tracking?
>
> **A2**: To perform annotations, we track the poses within a very small interval, i.e. 50 frames. We ask the mechanical turkers to check the tracking results, correct the wrong ones, and ensure most annotations are correct. We are not able to provide an estimation but we are confident about the quality of our annotations for test data.
>
> ---
>
> **Q3**: Is the instance segmentation network pre-trained (and if yes, how / on which data), or trained along with the rest of the network in an end-to-end manner? For the latter case, I'd expect it to be rather unstable at the beginning of the training.
>
> **A3**: We use the off-shelf instance segmentation model pretrained on the COCO dataset. Since the objects are relatively centered in our dataset, most segmentation results are accurate and we show some examples in our supplementary material pdf (Figure 1).
>
> ---

---

### Official Review · Reviewer_1VmD · 2022-07-15

**Rating:** 2
**Confidence:** 5
**Soundness:** 1 poor
**Presentation:** 1 poor
**Contribution:** 2 fair

**Summary:**

The paper proposes a method that utilizing unlabeled data for training to enable the proposed category-level 6Dpose estimation algorithm generalised to new scenarios. A new dataset is collected for implementing the idea. Comparisonal experiments were conducted on the REAL275 dataset and the proposed Wild6D dataset. The proposed algorithm has certain advantages over the listed algorithms.


**Questions:**

please check the strengths and weaknesses.

**Limitations:**

I didn't see any negative societal impact bring by this paper.

**Strengths And Weaknesses:**

Strengths:

1. The new method is able to utlizing the unlabeled real-scene data.

2. The authors give a large RGB-D real dataset and annotate the testing dataset, which may be useful for certain tasks.

3. The authors promise to make the code publicly available

Weaknesses and Questions:

1. The value of IOU-0.5 of CPS++ reported in Table 5 is way much lower than the value reported in the paper [1], please explain this inconsistency.

2. The author simply compared the performance of RePoNet with CPS++ when presenting the effectiveness of RePoNet on processing unlabeled dataset. While many SOTA unsupervised algorithms, for example, UDA-COPE [1], behaves better than RePoNet on the REAL275 dataset without any real annotations. It is necessary to compare more relevant SOTA methods for a fair comparison.

3. In Table 6, the CASS, Shape-Prior, DualPoseNet are all trained on the CAMERA75+REAL275, while RePoNet-syn and RePoNet-semi are trained either on CAMERA75 or CAMERA75+Wild6D. The author validated all these algorithms on the Wild6D and then emphasize the superiority of RePoNet. The lack of consistency on training dataset makes this comparison, again, unfair and invalid.

4. For line 327, the authors stated that 'FS-Net does not release the model, we cannot experiment on it.' I am not fully convinced by the explanation. Although the FS-Net doesn't release the pre-trained model, the code is available on GitHub (https://github.com/DC1991/FS_Net). Given the prominent performance of FS-Net on IOU-0.5, it is worth to try, which I believe it is also possible, to experiment with FS-Net on Wild6D dataset based on the published code.

[1] 'UDA-COPE: Unsupervised Domain Adaptation for Category-level Object Pose Estimation', CVPR2022

---

> ### Author Response · Authors · 2022-08-02
> **Thank you and response to 1VmD**
>
> We thank the reviewer for the comments. We address each concern in the following. We have also highlighted the new contents in our revised paper with the color red.
>
> ---
>
> **Q1:** The value of IOU-0.5 of CPS++ reported in Table 5 is way much lower than the value reported in the paper.
>
> **A1:** We report the results from Table5 (Top part) of the CPS++ paper, which is without using ICP for post-processing. This setting is consistent with our approach where we did not apply ICP post-processing for fair comparisons. We assume the reviewer is referring to the CPS++ results with ICP, which is **still worse than our approach** as shown below. We have also added this line of results in our paper for completeness.
>
> | Method | IOU\@0.5 | 5cm, 5 degree| 10cm, 5 degree |
> |----------|--------------|----------|-------------|
> | CPS++ w ICP | 72.8 | 25.2 | <58.6 |
> | Ours | **76.0** | **33.9** | **63.0**|
>
> ---
>
>
> **Q2**: While many SOTA unsupervised algorithms, for example, UDA-COPE [1], behave better than RePoNet on the REAL275 dataset without any real annotations. It is necessary to compare more relevant SOTA methods for a fair comparison.
>
> **A2**: We would like to kindly remind the reviewer that UDA-COPE is just published in CVPR2022 (code not available), after the NeruIPS deadline. It should be considered “concurrent to NeurIPS submissions” according to the NeurIPS policy. Our focus in this paper is on in-the-wild pose estimation. Although we perform ablation on REAL275, our goal and main contributions are both the new Wild6D dataset and the performance there. Finally, our performance is comparable to UDA-COPE on REAL275: For example, on the metric of 5 degree, 2cm, our method achieves 30.7% better than UDA-COPE with 30.4%, on the metric of 5 degree, 5cm, our method achieves 33.9% and UDA-COPE achieves 34.8%. We have added the result of UDA-COPE in our revised paper.
>
> ---
>
>
> **Q3**: The author validated all these algorithms on the Wild6D and then emphasized the superiority of RePoNet. The lack of consistency in the training dataset makes this comparison, again, unfair and invalid.
>
> **A3**: CASS, Shape-Prior, DualPoseNet are not trained with unlabeled Wild6D because they are not able to. It is nontrivial to leverage unlabeled data for 6D pose estimation, which is exactly one of the main contributions of our work. We also emphasize that the previous methods utilize annotations of real-world data while ours do not. We argue it is unreasonable to ask for aligning training data settings in this particular case since our goal is to prove our method of being able to use more unlabeled data (Wild6D) is better than previous approaches using the labeled dataset (REAL275).
>
> ---
>
>
> **Q4**:No comparison with FS-Net.
>
> **A4**: As mentioned by the reviewer that FS-Net has not released the model, we have tried to train FS-Net with the released code following their instructions, however, it does not work as same as mentioned in their paper. Multiple other users from Github have also reported that the performance cannot be reproduced from the released code: https://github.com/DC1991/FS_Net/issues/14 We kindly ask the reviewer to look into the issues brought up in the Github repo, most of the critical ones have not been addressed by the authors.
>
> We would also like to mention that the outstanding performance of IOU in FS-Net comes from a better tuned 2D object detector with YOLOv3 for pre-processing before pose estimation, thus the comparisons are also unfair.
>
> ---

---

### Author Response · Authors · 2022-08-05
**Following up on the post-rebuttal discussion**

Dear ACs and Reviewers,

Thank you so much again for the detailed feedback. We have reached half way of the author-reviewer discussion period. However, there are no responses yet to our replies.

Please do not hesitate to let us know if there are any further information and clarification we can provide. We hope to deliver all the information in time before the deadline.

Thank you!
Paper3142 Authors

---

### Meta-Review · Area_Chair_fgFy · 2022-08-28

**Recommendation:** Accept
**Confidence:** Certain

**Metareview:**

This paper received 4 reviews with the following scores: SR - BR - WA - A. The reviewers acknowledged importance of the addressed problem, the dataset contribution, clear presentation, and a meaningful approach with solid empirical performance. Main disagreements were around comparisons with existing methods (some published at CVPR'22), and fairness of training setups (supervised-only vs augmentation with real data).
The AC confirms that per NeurIPS policies the lack of comparisons with CVPR'22 publications indeed cannot be a basis for rejection.
However, looking at the additional results in Table 5 (including the CVPR 2022 paper) it looks like methods have actually been compared on both datasets using the various training regimes. Given that and that the remaining concerns of the reviewers were largely addressed in the rebuttal, both the AC and the SAC recommend acceptance.

**Award:**

No

---

### Decision · Program_Chairs · 2022-09-14

Accept